# Transcription Factor and miRNA Interplays Can Manifest the Survival of ccRCC Patients

**DOI:** 10.3390/cancers11111668

**Published:** 2019-10-28

**Authors:** Shijie Qin, Xuejia Shi, Canbiao Wang, Ping Jin, Fei Ma

**Affiliations:** Laboratory for Comparative Genomics and Bioinformatics, College of Life Science, Nanjing Normal University, Nanjing 210046, China; 191201005@stu.njnu.edu.cn (S.Q.); 181202063@stu.njnu.edu.cn (X.S.); 191202061@stu.njnu.edu.cn (C.W.)

**Keywords:** ccRCC, prognostic biomarker, miRNA, transcription factor, interplay

## Abstract

Clear cell renal cell carcinoma (ccRCC) still remains a higher mortality rate in worldwide. Obtaining promising biomakers is very crucial for improving the diagnosis and prognosis of ccRCC patients. Herein, we firstly identified eight potentially prognostic miRNAs (hsa-miR-144-5p, hsa-miR-223-3p, hsa-miR-365b-3p, hsa-miR-3613-5p, hsa-miR-9-5p, hsa-miR-183-5p, hsa-miR-335-3p, hsa-miR-1269a). Secondly, we found that a signature containing these eight miRNAs showed obviously superior to a single miRNA in the prognostic effect and credibility for predicting the survival of ccRCC patients. Thirdly, we discovered that twenty-two transcription factors (TFs) interact with these eight miRNAs, and a signature combining nine TFs (*TFAP2A*, *KLF5*, *IRF1*, *RUNX1*, *RARA*, *GATA3*, *IKZF1*, *POU2F2*, and *FOXM1*) could promote the prognosis of ccRCC patients. Finally, we further identified eleven genes (hsa-miR-365b-3p, hsa-miR-223-3p, hsa-miR-1269a, hsa-miR-144-5p, hsa-miR-183-5p, hsa-miR-335-3p, *TFAP2A*, *KLF5*, *IRF1*, *MYC*, *IKZF1*) that could combine as a signature to improve the prognosis effect of ccRCC patients, which distinctly outperformed the eight-miRNA signature and the nine-TF signature. Overall, we identified several new prognosis factors for ccRCC, and revealed a potential mechanism that TFs and miRNAs interplay cooperatively or oppositely regulate a certain number of tumor suppressors, driver genes, and oncogenes to facilitate the survival of ccRCC patients.

## 1. Introduction

Clear cell renal cell carcinoma (ccRCC) is the most common malignant tumor subtype of kidney cancer [1], which still remains a higher mortality rate in worldwide [2]. At present, the main treatment method on ccRCC patients is early resection, but its curative effect and prognosis are not very good for these terminally ccRCC patients [3]. Currently, above 30–50% of ccRCC patients have missed the best surgical opportunity due to the lack of early clinical symptoms [3]. Therefore, acquiring new molecular biomarkers not only are urgently needed for establishing clinically stratifying system to improve the diagnostic efficiency of ccRCC patients, but are of great clinical value for effectively improving the management and treatment strategy of ccRCC patients. 

MicroRNAs (miRNAs), a class of small noncoding RNAs with about 22 nucleotides, can negatively regulate gene expression at the post-transcriptional level by influencing the mRNA degradation and/or translational efficiency involved in manifold biological processes [4,5,6,7]. In recent years, many studies have revealed that miRNAs not only can act as oncogenes or tumor suppressor genes involve in tumorigenesis and progressions of various cancers [8], but can serve as valuable boimarkers for the detection and prognosis of cancer patients [9,10,11,12,13,14,15]. Thus, the current study has been focused on finding novel miRNAs as effective prognostic predictors for the overall survival of cancer patients [16,17,18,19,20]. 

At present, some studies have suggested that a lot of miRNAs could act as oncogenic miRNAs or tumor suppressor miRNAs to participate in the tumorigenesis and progressions of ccRCC, and they might serve as diagnosic and prognosic biomarkers for ccRCC patients [21,22,23,24]. However, several studies have shown that some miRNAs may play complete contradictor roles in diagnosic and prognosic effects for ccRCC, such as miR-99a [25,26], miR-106a [27,28], miR-125b [21,29], miR-144 [30,31], miR-203 [32,33], and miR-378 [34,35], which might greatly limit their applications into the clinical diagnosis and prognosis for ccRCC patients. Therefore, we should further study the functional roles of miRNAs as potentially diagnostic and prognostic biomarkers for ccRCC, whilst it is also very necessary for expanding the screening of new reliable miRNAs as diagnostic and prognostic biomarkers for ccRCC. 

Nowadays, a number of studies are mainly focused on the fuctional mechanism of miRNAs that regulate their targets to cause the occurrence and development of ccRCC. However, how are miRNAs themselves regulated in the occurrence of ccRCC, which is still largely unknown. Intriguingly, currently, many studies have shown that transcription factors (TFs) can regulate miRNA expressions, and miRNAs may also regulate TF expressions in gene regulatory networks [36,37,38,39], and TFs and miRNAs interplay could precisely modulate gene expressions to maintain cell homeostasis [37,40,41]. Therefore, considering the complexity of TFs and miRNAs interplay mediating gene expression, substantial works should be further made in elucidating the mechanism that TFs and miRNAs interplay drives the occurrence and development of ccRCC, and finding TFs and miRNAs as novel diagnostic and prognostic biomarkers in the survival of ccRCC patients.

Considering that miRNAs often carry out their functions through fine-tuning the expression of their target genes, and multiple miRNAs can also synergistically or antagonistically regulate one or more target genes to control the strength and duration of cell response [42,43]. Here, we firstly screened eight potentially prognostic miRNAs based on RNA-seq and clinical information from the TCGA database. We next combined the eight miRNAs as an integrative prognostic predictor to evaluate the prognostic efficiency. This result showed that the prognostic efficiency and credibility of the eight-miRNA signature significantly outperformed a single miRNA, which implies that the synergistical regulation of miRNAs plays key roles in the tumorigenesis and progression of ccRCC. Subsequently, we utilized target prediction software and KEGG enrichment analysis to find that these eight miRNAs mainly control a certain number of oncogenic and onco-suppressive genes from some crucially cancer-related pathways improving the survival of ccRCC patients. Additionally, we found that twenty-two TFs could interact with eight miRNAs based on deepCAGE, TransmiR v2.0, and MirWalk3.0 database [44,45,46]. To further reveal the possible molecular mechanism that TFs and miRNAs interplay facilitates the survival of ccRCC patients, we further constructed a interplay network of TFs and miRNAs and the network analysis revealed that the interplay between twenty-two TFs and eight miRNAs could synergistically control the expression of oncogenes, driver genes, and tumor suppressor genes to involve in the regulation of tumorigenesis and progression of ccRCC. Finally, we performed cox regression analysis to identify eleven genes as a eleven-gene signature, including six miRNAs and five TFs and the prognostic effect and the credibility of the eleven-gene signature also obviously outperformed the eight-miRNA signature and the nine-TF signature. Taken together, our findings not only revealed a novel possible mechanism that TFs and miRNAs interplay could regulate cooperatively oncogenes, driver genes, and tumor suppressor genes to facilitate the survival of ccRCC patients, but also identified some new potential prognostic factors and therapeutic targets for ccRCC patients. 

## 2. Results

### 2.1. Identification of miRNAs as Potential Prognostic Biomarkers

In this work, we found 110 differentially expressed miRNAs between 480 ccRCC tissues and 68 paracancerous tissues, including 50 up-regulated and 60 down-regulated miRNAs (Appendix A). Here, to identify these prognostic miRNAs for predicting the overall survival in ccRCC patients, we grouped 480 patients with at least 90 days into high- and low-expression groups according to the median expression level of each of 110 differentially expressed miRNAs. We next performed survival analysis to find that 37 miRNAs might be associated with the survival of ccRCC patients (*p*-value < 0.05) (Appendix A). Next, we performed univariate analysis and screened these top 21 miRNAs with a *p*-value < 0.001 (Figure 1, Appendix A) for multivariate stepwise cox regression analysis to further determine independently prognostic miRNA biomarkers for ccRCC patients. We finally obtained eight potential prognostic miRNAs, including four down-regulated miRNAs (hsa-miR-9-5p, hsa-miR-1269a, hsa-miR-183-5p, hsa-miR-335-3p) and four up-regulated miRNAs (hsa-miR-365b-3p, hsa-miR-223-3p, hsa-miR-144-5p, hsa-miR-3613-5p), which might be involved in the survival of ccRCC patients. We divided 480 ccRCC patients into high-risk and low-risk groups according to the median univariate cox risk score of each of these eight miRNAs to further detect the association between these identified eight miRNAs and the overall survival of ccRCC patients. Kaplan-Meier survival analysis and log-rank test indicated that, in all eight independent miRNA cohorts, ccRCC patients with high-risk groups exhibited the overall survival more badly than low-risk groups (all cohorts *p*-value < 0.01) (Appendix A). Interestingly, the range of the AUC value for eight miRNAs was about 0.6~0.7 (Appendix A), which indicated that the established prognosis model has a very good prognosis effect. Overall, these findings suggested that any of eight miRNAs might act as a possible prognostic biomarker for the survival of ccRCC patients.

### 2.2. The Exprssion Level of Eight Prognostic miRNAs is Associated with the Survival of ccRCC Patients

To explore whether the eight miRNAs could be used as potential diagnostic biomarkers for distinguishing patients with ccRCC from controls. Here, we used survival curves to assess the association between the expression level of these eight miRNAs and the overall survival of ccRCC patients. We divided ccRCC patients into high-expression and low-expression miRNA groups according to the median expression level of each of these eight miRNAs. This result indicated that, in all eight independent miRNA cohorts, ccRCC patients with high-expression miRNA groups exhibited a worse overall survival rate than the low-expression miRNA groups, except for hsa-miR-144-5p (Figure 2). These results seemed to indicate that these highly expressed hsa-miR-9-5p, hsa-miR-1269a, hsa-miR-183-5p, hsa-miR-335-3p, hsa-miR-365b-3p, hsa-miR-223-3p, and hsa-miR-3613-5p might be associated with a poor prognosis for ccRCC patients. It is noteworthy that the highly expressed hsa-miR-144-5p was associated with better overall survival (Figure 2), which suggested that hsa-miR-144-5p should be a good prognostic factor for ccRCC patients. In addition, we further calculated the association between the expression level of eight miRNAs and patient’s clinical diagnostic factors, respectively. Our results showed that eight prognostic miRNAs were significantly associated with T stage, M stage, G stage, and pathologic stage (Appendix A, Appendix A), implying that these eight miRNAs might be involved in tumorigenesis and progression of ccRCC and could be served as prognostic biomarkers for the survival of ccRCC patients. 

### 2.3. Prognostic Value of Combined Eight miRNAs as a Signature in ccRCC Patients

Considering multiple miRNAs could synergistically or antagonistically regulate one or more target genes to control cell fate. Herein, we further combined these eight miRNAs as an integrative prognostic predictor. The 480 patients were divided into low-risk group and high-risk group and then subjected to survival analysis. Our results showed that there was a significant difference in the overall survival between the two risk groups, and the high-risk group had more worse overall survival than the low-risk group (*p* < 0.0001, Figure 3A). In addition, the ROC curve based on the eight-miRNA signature also, respectively, showed an average 3, 5, and 10 year AUC for 0.762, 0.747, and 0.746 (Figure 3B). Interestingly, the concordance index (0.7305) of the combined prognostic model of the eight-miRNA signature was higher than that of all single miRNA (Appendix A), whereas the Akaike information criterion (1622.2491) of the combined prognostic model of the eight-miRNA signature was lower than that of all single miRNA (Appendix A), which indicated that the prognostic effect and credibility of the eight-miRNA signature were clearly superior to all single miRNA. These findings suggested that the eight-miRNA signature could act as a prognostic biomarker for promoting the survival of ccRCC patients. 

Cox proportional hazard regression analysis was further used to characterize the impact of various clinical factors on the overall survival of ccRCC patients (Table 1). Age, gender, tumor size, metastasis, pathologic stage, neoplasm histologic grade, and the combined eight miRNAs signature were coded as continuous variables. As shown in Table 1, the univariate analysis showed that all factors, except for gender, might act as prognostic indicators for ccRCC patients. However, the multivariate analysis indicated that only age and the eight-miRNA signature can be used as independent prognostic indicators for ccRCC patients. This result revealed that the eight-miRNA signature could not only could serve as an independent prognostic factor for overall survival of ccRCC patients, but also act as an effective risk stratification indicator for ccRCC patient diagnosis. 

### 2.4. Function Roles of Eight Prognostic miRNAs in ccRCC 

To reveal the functional role of eight prognostic miRNAs in ccRCC, we first identified 3672 differentially expressed genes between 480 ccRCC tissues and 68 paracancerous tissues, including 2057 up-regulated genes (of which 116 transcription factors) and 1012 down-regulated genes (of which 61 transcription factors) (Appendix A). Next, we obtained 357 down-regulated targets of 4 up-regulated miRNAs (hsa-miR-365b-3p, hsa-miR-223-3p, hsa-miR-144-5p, hsa-miR-3613-5p) and 1012 up-regulated targets of 4 down-regulated miRNAs (hsa-miR-9-5p, hsa-miR-1269a, hsa-miR-183-5p, hsa-miR-335-3p). 

To elucidate the function of these target genes of these eight miRNAs in ccRCC, we further performed KEGG pathway analysis using clusterProfiler R package. We found that these significantly up-regulated target genes of four down-regulated miRNAs were widely involved in cancer-related signaling pathways, such as MAPK, Ras, NF-kappa B, Chemokine and Cytokine-cytokine receptor (Appendix A). These results suggested that the interplay among multiple signaling pathways might synergistically mediate the occurrence and development of ccRCC. 

We further picked out these up-regulated target genes of four down-regulated miRNAs from these main cancer signaling pathways to construct the miRNA-gene regulation network (Figure 4A). As shown in Figure 4A, some up-regulated target genes were regulated by more than one down-regulated miRNA, implying that the cooperative regulation of multiple miRNAs might play key roles in the initiation and progression of ccRCC. In addition, the protein-protein interaction (PPI) network was also constructed for these up-regulated target genes of four down-regulated miRNAs using the STRING database, which demonstrated a close interaction within these target genes (Figure 4B). Here, a node with ≥ 20 degrees is defined as a hub gene, thus we found 30 hub genes (Appendix A). We next used multivariate cox regression analysis for 30 hub genes to further select prognosis-related genes for ccRCC patients. We found ten potential prognostic genes, including eight tumor suppressor and/or driver genes (*VEGFA*, *CCND1*, *BAX*, *IL7R*, *SHC1*, *FLT1*, *IL7,* and *JAK3*) [47,48,49,50,51], as well as two chemokines (*CXCL9* and *CXCL10*) [50]. Additionally, we combined the above ten hub genes as a signature to perform survival analysis. The result indicated that the low-risk group has a better overall survival rate than the high-risk group (*p* < 0.0001, Figure 4C), whilst the ROC curve also demonstrated that the ten-hub gene signature had a better prognostic effect and credibility for ccRCC patients with an average 3, 5 and 10 year AUC for 0.728, 0.751 and 0.796, respectively (Figure 4D). Taken together, our present findings suggested a possible molecular mechanism that the down-regulated expressed hsa-miR-9-5p, hsa-miR-1269a, hsa-miR-183-5p, and hsa-miR-335-3p might cooperatively up-regulate the expression level of numerous tumor suppressor and/or driver genes from some cancer-related pathways to improve the overall survival of ccRCC patients.

Compared to four down-regulated miRNAs, these down-regulated target genes of four up-regulated miRNAs were less enriched in cancer-associated signaling pathways (Appendix A). However, we further analyzed these target genes of three up-regulated miRNAs (hsa-miR-365b-3p, hsa-miR-223-3p, hsa-miR-3613-5p), finding that many genes of their target genes are involved in some cancer-associated signaling pathways (Figure 5A). Therefore, here, we reused multivariate cox regression analysis to screen potential prognostic genes for ccRCC patients. The result showed that five genes (*PRKCA*, *ADORA1*, *PPARGC1A*, *KL*, *GNG7*) could be identified as potentially prognostic factors, and they have also been suggested as tumor suppressor genes [49]. Interestingly, the survival analysis indicated that the five-gene signature could significantly stratify ccRCC patients into a high- and low-risk group (*p* < 0.0001, Figure 5B), and the AUC value of an average 3, 5 and 10 year is 0.696, 0.698, and 0.708, respectively (Figure 5C). Overall, we proposed a possibly functional mechanism that three highly expressed miRNAs (hsa-miR-365b-3p, hsa-miR-223-3p, and hsa-miR-3613-5p) might synergistically down-regulate the expression of many tumor suppressor genes to decrease the survival of ccRCC patients. 

Interestingly, our above results have indicated that the highly expressed hsa-miR-144-5p could act as a good prognostic factor for ccRCC patients. How does the highly expressed hsa-miR-144-5p facilitate the survival of ccRCC patients? Thus, we carried out a in-depth analysis for these target genes of hsa-miR-144-5p. Intriguingly, we found that hsa-miR-144-5p could regulate 55 genes, and, in particular, twelve genes of them have been reported as oncogenes or driver genes (Appendix A) [49]. Thus, we further used multivariate cox regression analysis for twelve oncogenes and driver genes to screen potential prognostic factors for ccRCC patients. Consequently, these five genes (*MAGI3*, *CDKL1*, *CDH1*, *PPM1K*, *MSI2*) could be served as potential prognostic factors. We further combined the five genes as an integrative prognostic predictor for survival analysis. These results showed that the high-risk group had a worse overall survival than the low-risk group (*p* < 0.0001, Figure 6A), and the AUC value of an average 3, 5 and 10 year is 0.659, 0.676 and 0.759, respectively, based on the ROC curve (Figure 6B). Collectively, our present findings implied that the highly expressed hsa-miR-144-5p might facilitate the overall survival of ccRCC patients through down-regulating the expression level of some certain oncogenes and/or driver genes. 

### 2.5. The Interplay Network Between Twenty-Two TFs and Eight miRNAs 

To explore the interplay between TFs and eight prognostic miRNAs, we further used TransmiR2.0 to predict these TFs that may regulate the eight prognostic miRNAs. Here, we found six down-regulated TFs (KLF5, SREBF2, TFAP2A, HIF1A, GATA3, and GATA2), which could regulate three down-regulated miRNAs (hsa-miR-9-5p, hsa-miR-183-5p, and hsa-miR-335-3p), but no TF was found for regulating hsa-miR-1269a (Appendix A). Whilst, 16 up-regulated TFs (CEBPA, E2F1, FOXM1, FLI1, HEY1, IKZF1, IRF1, MEF2C, MYC, POU2F2, POU5F1, PRDM1, RARA, RUNX1, RUNX3, and TCF4) were also found to regulate four up-regulated miRNAs (hsa-miR-365b-3p, hsa-miR-223-3p, hsa-miR-144-5p, hsa-miR-3613-5p) (Appendix A). Furthermore, we also predicted these twenty-two TFs regulated by the eight prognostic miRNAs. These detailed TF-miRNA regulatory pairs were outlined in the Appendix A. Based on the TF-miRNA pairs, we further constructed a TF-miRNA interplay network (Figure 7), which showed that these up-regulated TFs, such as MYC, IKZF1, and IRF1 could up-regulate the expression level of hsa-miR-223-3p and hsa-miR-365b-3p to inhibit the expression of some TFs, such as GATA2 and SREBF2, then reducing the expression of hsa-miR-183-5p and hsa-miR-335-3p to up-regulate tumor suppressor gene expression (Figure 7). Whilst these down-regulated TFs, such as KLF5, SREBF2, TFAP2A, HIF1A, GATA3, and GATA2, could also down-regulate the expression level of hsa-miR-9-5p, hsa-miR-183-5p, and hsa-miR-335-3p to up-regulate their target TF expression, such as E2F1, RUNX1, and RUNX3, and then up-regulated the expression of hsa-miR-365b-3p, hsa-miR-223-3p, hsa-miR-144-5p, and hsa-miR-3613-5p to down-regulate the expression of some tumor suppressor genes or oncogenes (Figure 7). Specially, TF and miRNA carry out opposing functions. Therefore, our study seemed to imply that the interplay between twenty-two TFs and eight prognostic miRNAs might precisely control the expression of oncogenes, driver genes, and tumor suppressor genes to facilitate the survival of ccRCC patients.

### 2.6. Prognostic Value of the Combined Nine TFs as a Signature in ccRCC Patients

Based on these above results, we further used multivariate cox regression analysis for these above twenty-two TFs to determine independent prognostic TFs for ccRCC patients. Herein, we identified nine potential prognostic TFs (*TFAP2A*, *KLF5*, *IRF1*, *RUNX1*, *RARA*, *GATA3*, *IKZF1*, *POU2F2,* and *FOXM1*) that could act as prognostic factors for ccRCC patients. We next combined these nine TFs as a signature to detect the prognostic effect for ccRCC patients. The 480 patients were divided into low-risk group and high-risk group and subjected to survival analysis. Our results showed that the high-risk group had worse overall survival than the low-risk group (*p* < 0.0001, Figure 8A). In addition, the ROC curve based on the nine-TF pool also, respectively, showed an average 3, 5, and 10 year AUC for 0.721, 0.748, and 0.780 (Figure 8B), indicating that the nine-TF signature had very good prognosis effect and credibility for the survival of ccRCC patients. These findings suggested that the nine-TF signature could serve as a prognostic biomarker for improving the survival of ccRCC patients.

### 2.7. Clinical Value of TFs and miRNAs Interplay as a Prognostic Signature in ccRCC Patients 

To further reveal the role of the interaction between TFs and miRNAs in the overall survival of ccRCC patients, we hypothesized that the interaction between TFs and miRNAs is likely to improve prognosis. Thus, we next performed multivariate analysis for above twenty-two TFs and eight miRNAs to identify prognostic TFs and miRNAs for ccRCC patients. We ultimately screened eleven potential prognostic factors, including hsa-miR-365b-3p, hsa-miR-223-3p, hsa-miR-1269a, hsa-miR-144-5p, hsa-miR-183-5p, hsa-miR-335-3p, *TFAP2A*, *KLF5*, *IRF1*, *MYC*, *IKZF1*. We further combined the above eleven genes as a signature for survival analysis. We found that the high-risk group had worse overall survival than the low-risk group (*p* < 0.0001, Figure 9A). Whilst the ROC curve based on the eleven-gene signature also, respectively, showed an average 3, 5, and 10 year AUC values for 0.777, 0.771, and 0.785 (Figure 10B). Strikingly, the concordance index (0.7552) of the combined prognostic model of the eleven-gene signature was more higher than that of the eight-miRNA signature (0.7305) and the nine-TF signature (0.7281), and the Akaike information criterion (1606.0516) of the combined prognostic model of the eleven-gene signature was lower than that of the eight-miRNA signature (1622.2941) and the nine-TF signature (1632.5955) (Appendix A), which indicated that the prognostic effect and the credibility of the eleven-gene signature were better than both the eight-miRNA signature and the nine-TF signature did. These findings suggested that the eleven-gene signature could act as a prognostic factor for the overall survival of ccRCC patients.

Here, we also further used Cox proportional hazard regression analysis to characterize the impact of various clinical factors on overall survival of ccRCC patients (Table 2). Age, gender, tumor-pathologic, metastasis pathologic, pathologic stage, neoplasm histologic grade, and the eleven-gene signature were coded as continuous variables. The univariate analysis showed that all factors, except for gender, might serve as prognostic indicators for ccRCC patients (Table 2). Notably, the multivariate analysis demonstrated that only age and the eleven-gene signature could be used as independent prognostic indicators for ccRCC patients (Table 2). Taken together, our present results revealed that the interaction between TFs and miRNAs might have very important effects on the overall survival of ccRCC patients, and the eleven-gene signature might serve as an independent factor to improve the prognosis for ccRCC patients. 

Overall, herein we propose a potential molecular mechanism that the interplay between TFs and miRNAs facilitates the overall survival of ccRCC patients (Figure 10). On the one hand, down-regulated expressed TFs could down-regulate some miRNA expression to further up-regulate tumor suppressor gene expression, whilst the down-regulated miRNAs could also up-regulate TF expression to further up-regulate the expression of tumor suppressor miRNA to down-regulate the expression of some oncogenes to improve prognosis for ccRCC. On the other hand, up-regulated TFs could up-regulate miRNA expression to inhibit the expression of some tumor suppressor genes, whilst the up-regulated miRNAs could also down-regulate the expression of some TFs to further down-regulate the expression of some miRNAs to up-regulate the expression of some tumor suppressor genes, which might promote ccRCC development. Taken together, our results seemed to reveal that the interplay between TFs and miRNAs might synergistically regulate the expression of a certain number of oncogenes, driver genes, and tumor suppressor genes to improve prognosis for ccRCC patients in transcriptional and post-transcriptional levels.

## 3. Discussion

At present, a number of miRNAs have been suggested as diagnosic and prognosic biomarkers for ccRCC patients, but, since a single miRNA mainly fine-tunes gene expression to execute its function, and the functional effect of a single miRNA is relatively weak, which might result in a lack of miRNA application into the clinical diagnosis and prognosis for ccRCC patients. Many studies have shown that miRNAs are more likely to regulate a certain number of gene expressions to control cell fate [52,53]. Remarkably, several studies have revealed that TFs can regulate miRNA expressions, whilst miRNAs may also regulate TF expressions, and TFs and miRNAs interplay can precisely modulate gene expression in transcriptional and post-transcriptional levels [36,45,54,55,56,57]. However, the regulatory landscape by miRNAs and TFs interplay is still largely unknown in the tumorigenesis and progression of ccRCC up to now. Especially, the interplay network of miRNA and TF has not yet been systemically studied in ccRCC. Therefore, in this current work, we have integratedly analyzed miRNA, TF, and mRNA profilings, and identified some potential diagnostic and prognostic factors participated in the survival of ccRCC patients, as well as revealed a possible molecular mechanism that miRNA and TF interplay can serve as an effective prognostic factor to facilitate the survival of ccRCC patients. 

In this study, we have identified eight potentially diagnostic and prognostic miRNAs that could significantly distinguish the survival and pathological stratification for ccRCC patients. Among them, four down-regulated miRNAs (hsa-miR-9-5p, hsa-miR-1269a, hsa-miR-183-5p, hsa-miR-335-3p) could serve as good prognostic factors for clinical application of ccRCC patients, conversely the three up-regulated hsa-miR-365b-3p, hsa-miR-223-3p and hsa-miR-3613-5p of them might been acted as poor prognostic factors for ccRCC patients (Figure 2). At present, a few studies have reported that hsa-miR-183-5p can promote canceration by targeting SRSF2 in renal cancer [58], miR-335 could inhibit the proliferation and invasion of clear renal cells through suppressing Bcl-w [59], hsa-miR-365b-3p is poorly associated with ccRCC patient survival [60] and hsa-miR-9-5p is associated with the development and risk of renal cancer recurrence [61]. Although studies on the function roles of the other three miRNAs in ccRCC are still less reported to date, some other studies have revealed that hsa-miR-1269a could function as an onco-miRNA in NSCLC via down-regulating its target SOX6 [62], as well as that miR-1269a could promote colorectal cancer (CRC) metastasis by targeting Smad7 and HOXD10 [63]. The highly expressed hsa-miR-3613 as a oncogene could inhibit apoptosis via the down-regulation of target *APAF1* in human neuroblastoma BE(2)-C cells, as well as serve as a potential prognostic biomarker for pancreatic adenocarcinoma [64]. Some recent studies have indicated that miR-223-3p could down-regulate aNHEJ expression to result in synthetic lethality in human BRCA1-deficient cancers [65], and also act as an oncogenic miRNA in colon cancer through regulating EMT and PRDM1 [66]. These previous studies have revealed that the high expression of the seven miRNAs could down-regulate the respective tumor suppressor genes or driver genes involved in different malignant tumor occurrences. Especially, the same miRNA might regulate different target genes involved in specific tumor formation, which suggests that the different roles of same miRNA might depend on different cell microenvironment and cancer types. Interestingly, our present study has indicated that the seven aforementioned miRNAs mainly regulate tumor suppressor and/or driver genes to involve in ccRCC. Thus, we suggest that the seven miRNAs could served as potentially diagnostic and prognostic factors that may be due to the four down-regulated miRNAs that could restore or elevate the expression level of a certain number of tumor suppressor or driver genes to improve the survival of ccRCC patients, whereas the three up-regulated miRNAs might decrease the expression level of many tumor suppressor genes to result in the worse survival of ccRCC patients. Particularly, our present findings have shown that the highly expressed hsa-miR-144-5p could reduce the expression level of multiple oncogenic genes to promote the survival for ccRCC prognosis. A previous study has also shown that the hsa-miR-144-5p could serve as a tumor-suppressor gene for inhibiting cell growth and arresting cells in the G1 phase in renal cancer [30], which is also agreement with our present result. Taken together, we urge that hsa-miR-223-3p, hsa-miR-365b-3p, hsa-miR-3613-5p, hsa-miR-9-5p, hsa-miR-183-5p, hsa-miR-335-3p, and hsa-miR-1269a could serve as oncogenic miRNAs, as well as the hsa-miR-144-5p could act as a tumor suppressor miRNA for ccRCC patient diagnosis, but it is still further experimental verification.

A single miRNA molecule is known to carry out its function through fine-tuning the expression of target genes [42,43]. Therefore, it is hard to say that the expression of a single gene regulated by a single miRNA could significantly impact the proliferation and migration of cancer cell. Here, our findings seem to imply that single miRNA might need to regulate the expression of a lot of genes involved in the progression of ccRCC, suggesting that a single miRNA might be more suitable for ccRCC diagnosis and prognosis than a single gene. Specially, multiple miRNAs prefer synergistically or antagonistically regulating one or more target genes to control the strength and duration of cell response [42,43]. Thus, the combined multi-miRNAs as diagnosic and prognosic factors might be more suitable for the clinical application of ccRCC patients than a single miRNA. In the present study, just as we wish the prognostic effect and the credibility of the eight-miRNA signature are clearly superior to a single miRNA (Figure 3, Appendix A), implying that the cooperative regulation of multi-miRNAs might play very important roles in tumorigenesis and progression of ccRCC. 

In the present work, we have found that eight prognostic miRNAs could interplay with twenty-two TFs. Many studies have revealed that the interplay between miRNA and TFs play key roles in establishing and maintaining cell phenotype [67,68,69]. Notably, in the gene regulatory network, TF and miRNA interplay could constitute positive or negative feedback loops to execute similar and opposing functions, which can precisely control the regulation of gene expression to reduce noise and maintain cell homeostasis [40,70,71]. Therefore, while considering the complexity of TFs and miRNAs interplay regulating gene expression, we further constructed an interplay network of TF-miRNA and propose a potential molecular mechanism that the interaction between TFs and miRNAs facilitates the survival of ccRCC patients (Figure 10). As shown in Figure 10, down-regulated transcription factors, such as KLF5 and GATA2, can down-regulate miR-183-3p and miR-1269a to up-regulated FAS and other tumor suppressor genes, whilst down-regulated miR-183-3p and miR-1269a can also up-regulate IKZF1 and IRF1 to down-regulated downstream targets. Additionally, up-regulated MYC and IKZF1 can activate miR-223-3p and miR-365b-3p to down-regulate RPS6KA6, DEPTOR, and other tumor suppressor genes, whilst miR-223-3p and miR-365b-3p can also down-regulate TFAP2A and GATA2 to down-regulate miR-183-5p and miR-1269a to up-regulated FAS and VEGFA. Remarkably, some previous reports showed that the transcription factor GATA2 could activate the expression of miR-194 to promote this distant metastasis of prostate cancer by inhibiting SOCS2 [72], and the transcription factor KLF5 could also promote the expression of miR-145, miR-124, and miR-183 by binding to their promoter involved in the progression of invasive pituitary adenoma [73], as well as the transcription factor TFAP2C could promote lung tumorigenesis and aggressiveness through it activating miR-183 and miR-33a-mediated cell cycle regulation [74]. Especially, the interplay of MYC and hsa-miR-144 has been reported in chronic myelogenous leukemia cell K562 [75], and the transcription factor E2F1 could also up-regulate miR-224/452 expressions to inhibit the expression of TXNIP to drive EMT in malignant melanoma [36], as well as miR-3188, as a tumour suppressor, could control the nasopharyngeal carcinoma proliferation and chemosensitivity through a mechanism where FOXO1 modulated a positive feedback loop of mTOR-p-PI3K/AKT-c-JUN [38]. These above results supported our conclusion that the interplays between transcription factors and miRNAs might play very important roles in the prognosis of ccRCC patients. Thus, based on the importance of TF and miRNA interplay in gene expression regulation, we ultimately screened six miRNAs (hsa-miR-365b-3p, hsa-miR-223-3p, hsa-miR-1269a, hsa-miR-144-5p, hsa-miR-183-5p, hsa-miR-335-3p) and five TFs (TFAP2A, KLF5, IRF1, MYC, IKZF1) as an integrative prognostic predictor. Interestingly, the prognostic effect and the credibility of the combined six miRNAs and five TFs signature are better than both the eight-miRNA signature and the nine-TF signature. The reason might be that not only TFs can regulate the expression of multiple target genes, including miRNAs, but also miRNAs can fine-tune multiple gene expressions, including TFs, as well as their closely coordinated regulations control cell homeostasis. This also suggests why TFs and miRNAs interplay is effective as a clinical prognostic factor for ccRCC patients. Of course, all of these regulatory pairs predicted by bioinformatics and data integration are still to be further verified experimentally. 

## 4. Materials and Method

### 4.1. Data Sources and Pre-Processing

All KIRC sample RNA-seq data of mRNA and miRNA isoform and corresponding clinical information were downloaded from the The Cancer Genome Atlas (TCGA) database (https://portal.gdc.cancer.gov/, Springer Netherlands, Bethesda, MD, USA). The samples were filtered based on survival days greater than three months and simultaneous possessing mRNA and miRNA expression data. Based on the AnimalTFDB3.0 (http://bioinfo.life.hust.edu.cn/AnimalTFDB/, Oxford University Press, Hubei, China.) and Ensemble (http://asia.ensembl.org/index.html, Oxford University Press, Cambridgeshire, UK.) annotations, we identified 19,780 coding genes, of which 1400 were transcription factors. The expression level of 2,104 miRNA matures was obtained after miRNA isoform alignment. These lower expressed genes (sum(cpm) < 1) were removed and these genes expressed in at least 50% of the sample were retained.

### 4.2. Differentially Expressed Gene Analysis

Difference gene expression analysis between all tumor and paracancerous tissues was performed while using the edgeR package with filter parameters |log2FC| > 1 and *p*.adjust < 0.05. Similarly, differentially expressed miRNA was screened by limma package with criteria: |log2FC| > 1, *p*.adjust < 0.05. In addition, the mRNA and miRNA expression profiles were further respectively converted to log2 (normalized value + 1) and log2 (RPKM + 1) to be used for the next operation.

### 4.3. Survival Analysis and Prognosis Model Establishment

The samples of ccRCC with OS > 90 days were selected for survival analysis. Firstly, according to the median expression of miRNAs, batch survival analysis was performed to screen out the significantly differentially expressed miRNAs that are associated with survival. Subsequently, the univariate Cox regression was used to further assess the miRNAs related to survival. Only those miRNAs with a *p*-value < 0.001 were selected as candidate biomarker miRNAs. Finally, these candidate miRNAs were subjected to multivariate cox regression to determine independent prognostic marker miRNAs and calculate the risk value constructing prediction model for each miRNA. Based on the above results, the time-dependent receiver operating characteristic (ROC) curve was drawn using the Survival ROC R package, and the classification model was evaluated according to the area under the curve (AUC). In addition, we also analyzed the concordance index (C-index) and the Akaike information criterion (AIC). The C-index represents the consistency of the probability of the actual occurrence of the outcome and the probability of the prediction, and the AIC represents a standard for measuring the goodness of statistical model fitting. The prognostic signature of individual calculated according to the risk values of each marker miRNA and combined miRNAs. Next, the patient were divided into high and low risk groups according to the median risk score, and the survival analysis curve was then performed to check significant difference of patients in two groups over time.

### 4.4. miRNA Target Prediction and Target Function Analysis

These target genes of prognostic miRNAs were predicted by integrating Mirwalk3.0 (http://mirwalk.umm.uni-heidelberg.de/, Public Library of Science, Mannheim, Germany.) and a negative correlation between miRNA and mRNA expression. The Kyoto Encyclopedia of Genes and Genomes (KEGG) analysis was performed using the clusterProfiler package [76] with the filtration standard: *p*.adjust < 0.05. The PPI network from target genes was derived from the STING database (https://string-db.org/, Oxford University Press, Zurich, Switzerland).

### 4.5. Prediction of Transcription Factors Regulating miRNAs

The TransmiR2.0 (http://www.cuilab.cn/transmir, Oxford University Press, Beijing, China.) has collected the human regulatory pair of TFs regulating miRNAs (TFs-miRNAs) based on accurate transcriptional start site (TSS) of miRNA and ChIP-seq sequencing as well as experimental validation. Combining the TFs-miRNAs regulatory pair with the expression positive correlation of miRNAs and TFs, and TFs that may regulate these biomarker miRNAs were screened.

### 4.6. Data Statistics and Visualization

All data analysis was performed using R software (version 3.5.1, R Core Team, Vienna, Austria). Cytoscape software [77] (version 3.6.1, Cold Spring Harbor Laboratory Press, Washington, WA, USA.) were used to visualize the network. The survival curve was plotted using Kaplan Meier function, and the difference significance was evaluated by log-rank test.

## 5. Conclusions

In this work, we have identified eight prognostic miRNAs. Among them, seven miRNAs (hsa-miR-223-3p, hsa-miR-365b-3p, hsa-miR-3613-5p hsa-miR-9-5p, hsa-miR-183-5p, hsa-miR-335-3p, hsa-miR-1269a) can serve as potential oncogenes, whereas hsa-miR-144-5p might act as a tumor suppressor gene for ccRCC diagnosis. In addition, the eleven-gene signature (hsa-miR-365b-3p, hsa-miR-223-3p, hsa-miR-1269a, hsa-miR-144-5p, hsa-miR-183-5p, hsa-miR-335-3p, *TFAP2A*, *KLF5*, *IRF1*, *MYC*, *IKZF1*) can sever as an effective prognostic predictor to significantly improve the overall survival of ccRCC patients. Especially, our study has revealed a possible molecular mechanism that TFs and miRNAs interplay can cooperatively regulate the expression of oncogenes, driver genes, and tumor suppressor genes to facilitate the survival of ccRCC patients. Thus, our findings not only provide a new insight into the mechanism that TFs and miRNAs interplay control the tumorigenesis and progression of ccRCC, but also identify several novel diagnostic and prognostic biomarkers as well as potential therapeutic targets that are very crucial for making individualized therapeutic strategies of ccRCC patients.

## Figures and Tables

**Figure 1 cancers-11-01668-f001:**
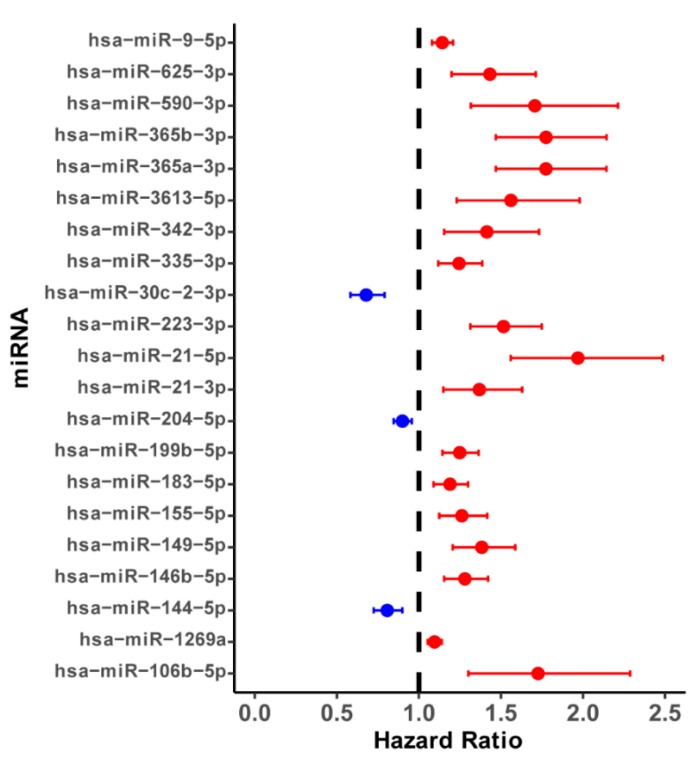
**The forest plots of hazard ratios (HR) of top 21 most significant survival associated miRNAs in ccRCC (*p*-value < 0.001).** Red represents the risk factors (HR > 1) and blue represents protective factors (HR < 1).

**Figure 2 cancers-11-01668-f002:**
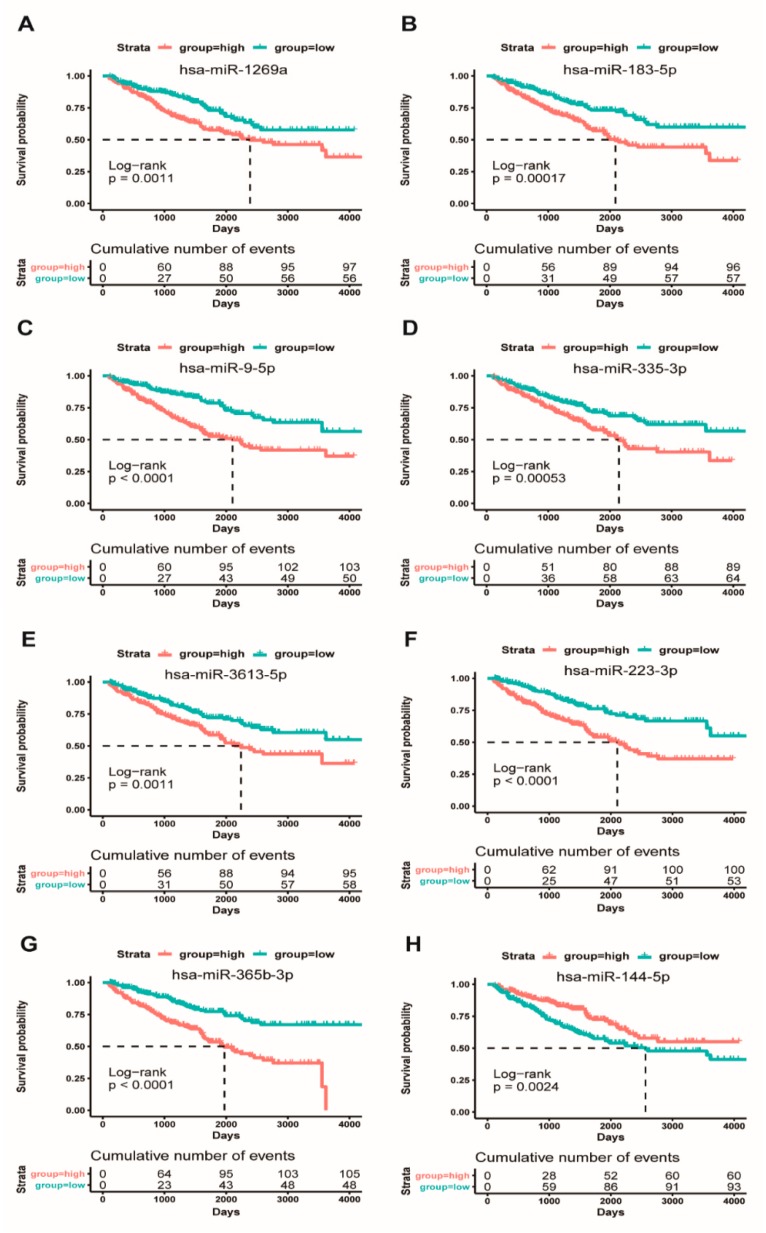
**Kaplan Meier survival based on the expression level of eight miRNAs.** Overall survival curves for high-expression and low-expression ccRCC patient cohorts. (**A**): hsa-miR-1269a; (**B**): hsa-miR-183-5p; (**C**): hsa-miR-9-5p; (**D**): hsa-miR-335-3p; (**E**): hsa-miR-3613-5p; (**F**): hsa-miR-223-3p; (**G**): hsa-miR-365b-3p; and, (**H**): hsa-miR-144-5p.

**Figure 3 cancers-11-01668-f003:**
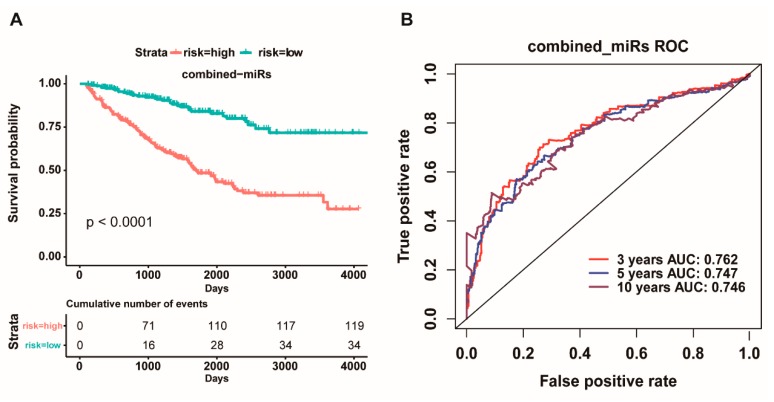
**Kaplan Meier survival and receiver operating characteristic (ROC) curves based on the riskscore of the eight-miRNA signature.** (**A**): Overall survival curves of high-risk and low-risk based on the eight-miRNA signature model. (**B**): Receiver operating characteristic (ROC) curves for high and low risk from the eight-miRNA signature model.

**Figure 4 cancers-11-01668-f004:**
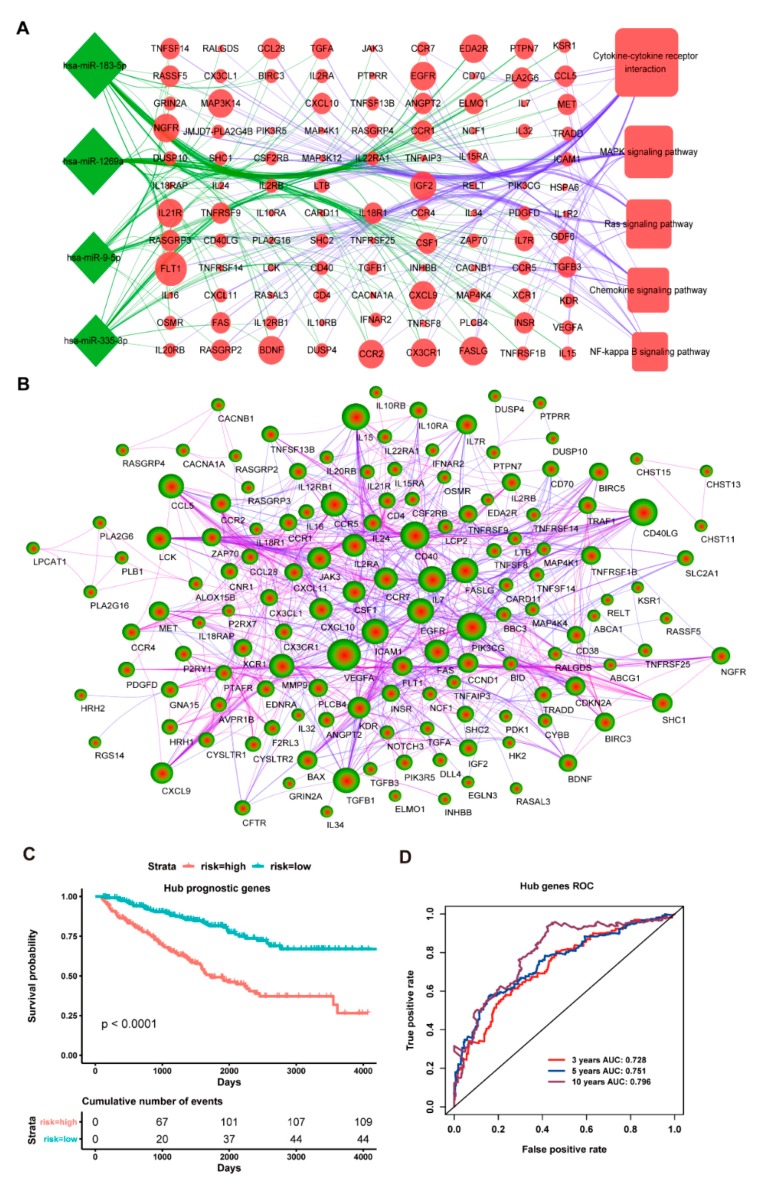
**The cancer signaling pathway and protein network analysis of these up-regulated targets and the prognostic model of the ten-hub gene signature**. (**A**): Target genes of four down-regulated miRNAs involve in the network map of cancer-related signaling pathways. The green represents the down-regulated expression and the red represents up-regulated expression; (**B**): Target protein interaction network of four down-regulated miRNAs. The blue line means low credibility and the purple line means high credibility; (**C**): Overall survival curves for high-risk and low-risk groups based on the ten-hub gene model; and, (**D**): ROC curves for high-risk and low-risk based on the ten-hub gene model.

**Figure 5 cancers-11-01668-f005:**
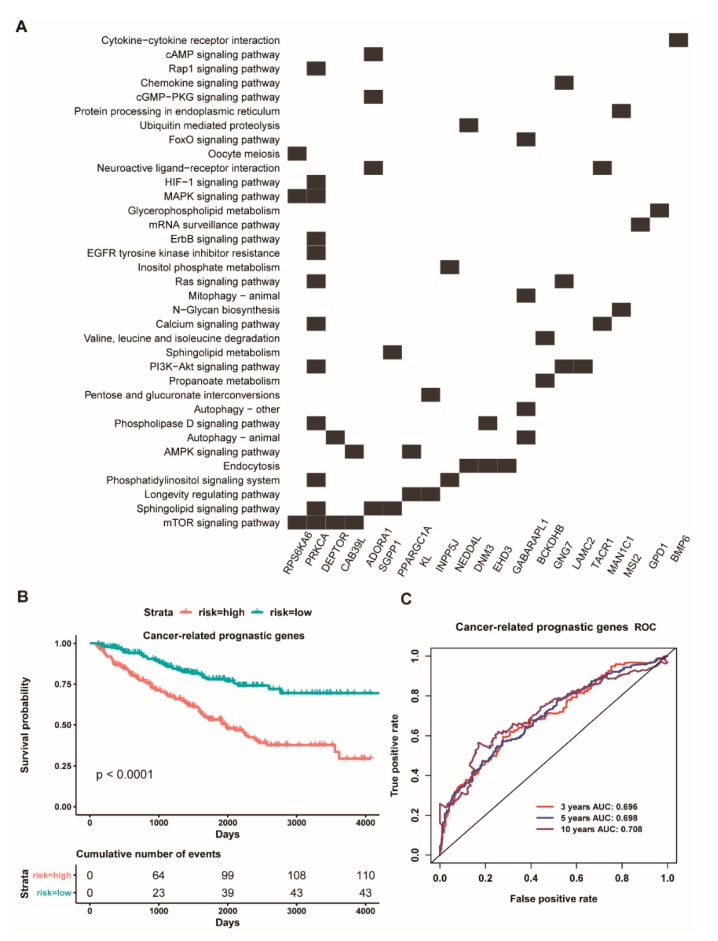
**The heat map of down-regulated targets of three up-regulated miRNAs involve in cancer-related signaling pathway and the prognostic model of the five-gene signature**. (**A**): The heat map of these down-regulated targets regulated by has-miR-3613-5p, has-miR-223-3p, and has-miR-365b-3p involved in cancer-related signaling pathways; (**B**): Overall survival curves for high-risk and low-risk groups based on the prognostic model of a five-gene signature; (**C**): ROC curves for high and low risk from the prognostic model of a five-gene signature.

**Figure 6 cancers-11-01668-f006:**
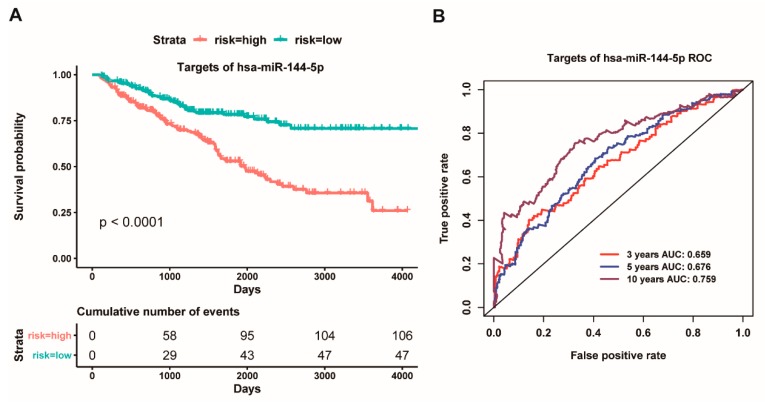
**Kaplan Meier survival and ROC curves based on the risk score of five prognostic targets of has-miR-144-5p.** (**A**): Overall survival curves for high-risk and low-risk groups based on the prognostic model of a five-gene signature; (**B**): ROC curves for high and low risk from the prognostic model of a five-gene signature.

**Figure 7 cancers-11-01668-f007:**
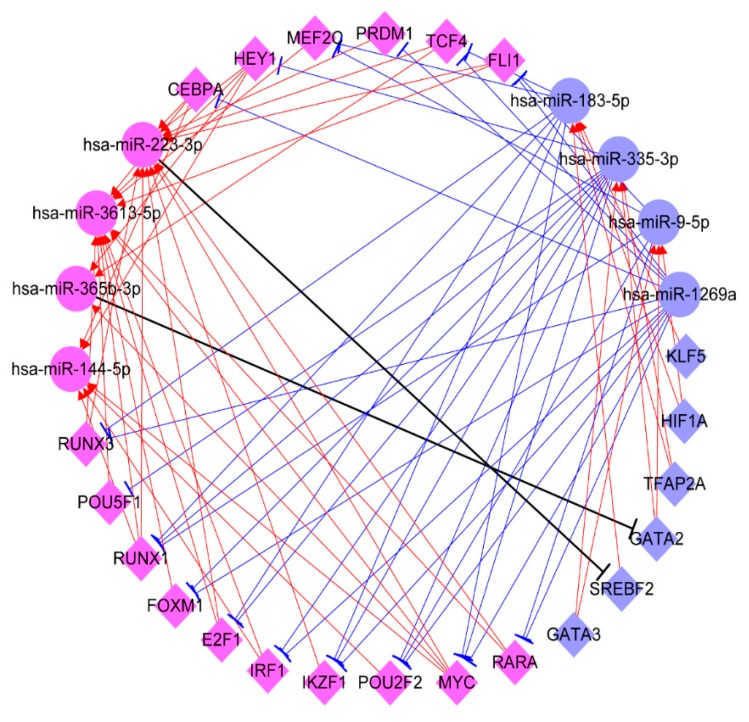
**The interplay network between twenty-two transcription factors and eight miRNAs.** The circle represents the miRNA and the diamond represents the transcription factor. The blue represents a down-regulation expression and the purple represents an up-regulation expression. The sharp arrow represents activation and the flat arrow represents inhibition.

**Figure 8 cancers-11-01668-f008:**
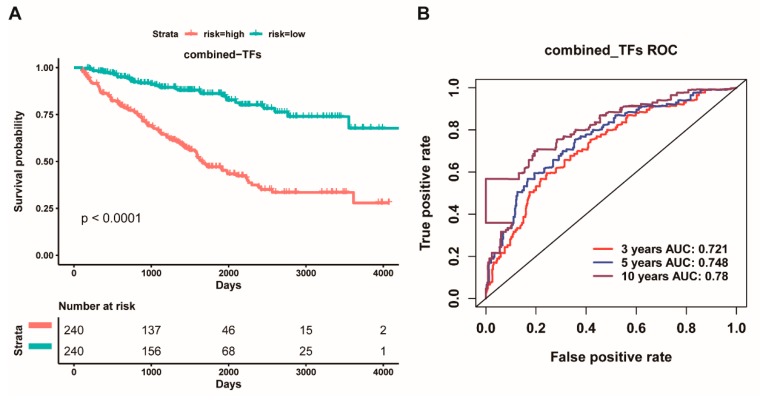
**Kaplan Meier survival and ROC curves based on the risk score of the combined nine transcription factors as a signature.** (**A**): Overall survival curves for high-risk and low-risk groups based on the prognostic model of a nine-TF signature; (**B**): ROC curves for high and low risk from the prognostic model of a nine-TF signature.

**Figure 9 cancers-11-01668-f009:**
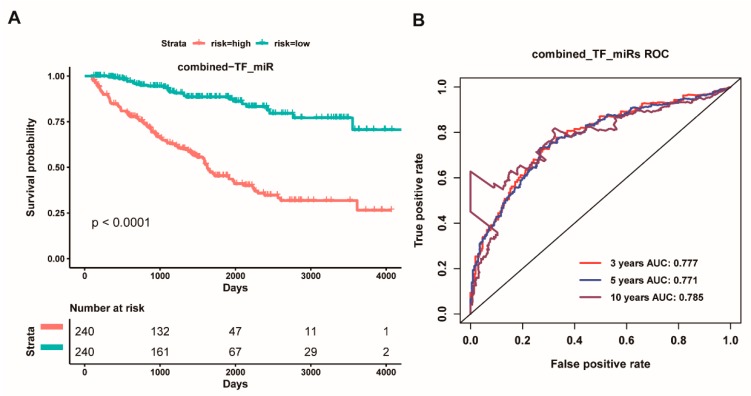
**Kaplan Meier survival and ROC curves based on the risk score of the combined five transcription factors and six miRNAs as a signature.** (**A**): Overall survival curves for high-risk and low-risk groups based on the prognostic model of an eleven-gene signature; and, (**B**): ROC curves for high and low risk from the prognostic model of an eleven-gene signature.

**Figure 10 cancers-11-01668-f010:**
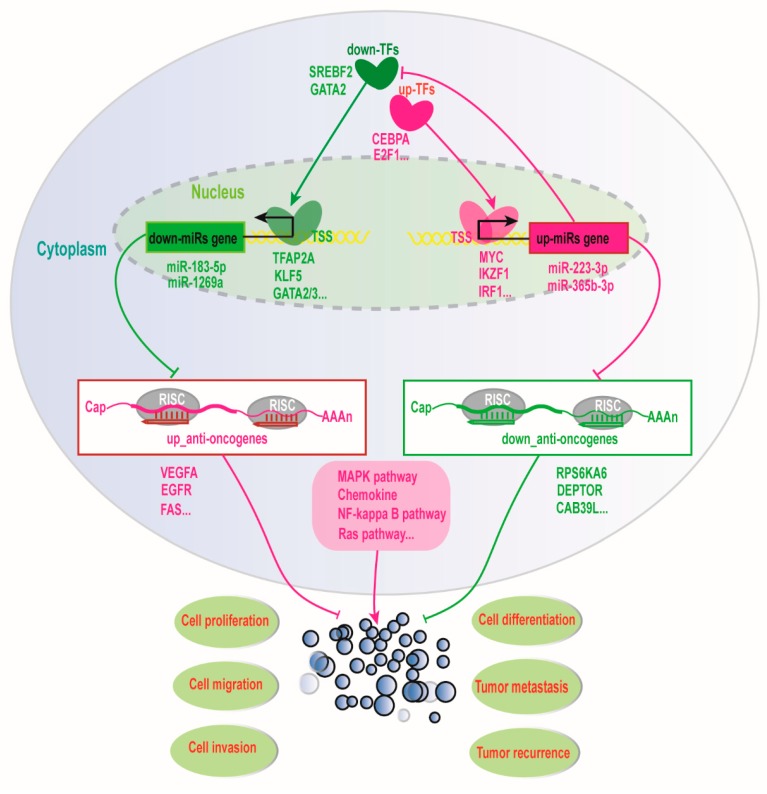
**The molecular mechanism of the interplay between transcription factors and miRNAs improving the prognosis of ccRCC patients.** The red represents an up-regulation of gene expression and the green represents down-regulation of gene expression. The sharp arrow represents activation and the flat arrow represents inhibition.

**Table 1 cancers-11-01668-t001:** Univariate and multivariate Cox regression analysis of the overall survival for clinical factors and risk of the combined eight prognostic miRNAs as a signature.

Variables	Univariate Analysis		Multivariate Analysis	
	Hazard Ratio (95% CI)	*p*-Value	Hazard Ratio (95% CI)	*p*-Value
Age	1.029 (1.014–1.043)	<0.001	1.028 (1.012–1.044)	<0.001
Gender	0.920 (0.660–1.284)	0.624	0.832 (0.590–1.172)	0.292
Tumor_pathologic_T	1.855 (1.556–2.211)	<0.001	0.816 (0.525–1.269)	0.367
Metastasis_pathologic_M	4.671 (3.369–6.475)	<0.001	1.514 (0.744–3.078)	0.253
Pathologic_stage_Stage	1.884 (1.634–2.172)	<0.001	1.605 (0.994–2.590)	0.053
Histologic_grade_G	2.238 (1.800–2.783)	<0.001	1.275 (0.996–1.633)	0.054
The eight-miRNA signature	4.009 (2.725–5.898)	<0.001	2.666 (1.768–4.020)	<0.001

Age, gender, tumor stage, metastasis pathologic, pathologic stage, histologic grade, and the eight-miRNA signature were coded as continuous variable. Specifically, pathologic stage was coded as I = 1, II = 2, III = 3, IV = 4. Tumor stage was coded as T1 = 1, T2 = 2, T3 = 3, T4 = 4. Histologic grade was coded as G1 = 1, G2 = 2, G3 = 3, G4 = 4.

**Table 2 cancers-11-01668-t002:** Univariate and multivariate Cox regression analysis of overall survival for clinical factors and risk of the combined five prognostic transcription factors and six miRNAs as a signature.

Variables	Univariate Analysis		Multivariate Analysis	
	Hazard Ratio (95% CI)	*p*-Value	Hazard Ratio (95% CI)	*p*-Value
Age	1.029 (1.014–1.043)	<0.001	1.028 (1.011–1.044)	<0.001
Gender	0.920 (0.6595–1.284)	0.624	0.949 (0.673–1.338)	0.765
Tumor_pathologic_T	1.855 (1.556–2.211)	<0.001	0.840 (0.539–1.311)	0.443
Metastasis_pathologic_M	4.671 (3.369–6.475)	<0.001	1.707 (0.830–3.513)	0.146
Pathologic_stage_Stage	1.884 (1.634–2.172)	<0.001	1.410 (0.865–2.300)	0.178
Histologic_grade_G	2.238 (1.800–2.783)	<0.001	1.420 (1.118–1.803)	0.765
The eleven-gene signature	4.349 (2.943–6.428)	<0.001	2.590 (1.706–3.930)	<0.001

Age, gender, tumor stage, metastasis pathologic, pathologic stage, histologic grade and the eleven-gene signature were coded as continuous variable. Specifically, pathologic stage was coded as I = 1, II = 2, III = 3, IV = 4. Tumor stage was coded as T1 = 1, T2 = 2, T3 = 3, T4 = 4. Histologic grade was coded as G1 = 1, G2 = 2, G3 = 3, G4 = 4.

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
