# Peer review of "Transcription Factor and miRNA Interplays Can Manifest the Survival of ccRCC Patients"

_cancers, 2019, doi:10.3390/cancers11111668_

Round 1

Reviewer 1 Report

In this manuscript (ID: cancers-616468), the authors identified eight differentially expressed miRNAs and nine transcription factors (TFs) that were potential in prognosis of the clear cell renal cell carcinoma (ccRCC) treatment. In particular, they further determined a combinational biomarker set of six miRNAs and five TFs that could substantially differentiate the survival status of ccRCC patients. Generally, this study has its values in prognosis of ccRCC treatment. It also provides clues to elucidate transcription factors and miRNAs in cancer development. Major concerns: 1. Page 2 Line 89-, whether the 110 differentially expressed miRNAs were consistently detectable in all 480 ccRCC and 68 paracancerous tissues? Were the paracancerous tissues significantly different from the cancer tissues in histopathology? 2. Page 5 Figure 3, the association analysis between miRNA expressions and clinical features was not so closely related the major content of this study. I suggested to move this section to the Supplementary Information, or even substantially shrink the content. 3. Page 12 Line 310-, the authors determined a combined signature set of 6 miRNAs and 5 TFs, which outperformed differentially expressed miRNAs or TFs in the ccRCC prognosis. How many molecules out of the eleven prognostic factors have been used in current commercial molecular panels for pan-cancer or ccRCC diagnosis? What are the functional connections between these selected miRNAs and TFs? 4. Page 14 Figure 11, it is very interesting mechanism map. However, discussion of the proposed mechanism in a more specified way (i.e. which miRNAs, TFs, target genes were involved? how were they involved? and how did they interplay with each other, and so on) is desired. 5. Generally, the authors need to reorganize the manuscript to make it focused, concise, and easy to read. BTW, I suggest the author to reconsider the article title.

Author Response

Response to Reviewer 1’s Comments

Point 1: Page 2 Line 89-, whether the 110 differentially expressed miRNAs were consistently detectable in all 480 ccRCC and 68 paracancerous tissues? Were the paracancerous tissues significantly different from the cancer tissues in histopathology?

Response 1: Thank you for your comments. In this work, these 110 differentially expressed miRNAs detected between 480 ccRCC and 68 paracancerous tissues are nearly consistent with the paired 68 ccRCC and 68 paracancerous tissues. Although the false positive still is present, it is within the acceptable statistical range (FDR < 0.05).  

Paracancerous tissues are actually very similar to normal tissues, but paracancerous tissues usually have significantly different from cancer tissues in histopathology. At present, most of the tumor research is to compare the difference between the adjacent tissues and the tumor tissues (eg.[1,2]). On the one hand, this can compare the difference between tumor and normal tissue. On the other hand, it can reduce the tissue sampling error caused by individual difference.

Point 2: Page 5 Figure 3, the association analysis between miRNA expressions and clinical features was not so closely related the major content of this study. I suggested to move this section to the Supplementary Information, or even substantially shrink the content.

Response 2: Thank you for your good suggestion. We have moved this section to the Supplementary Information in this revised manuscript.

Point 3: Page 12 Line 310-, the authors determined a combined signature set of 6 miRNAs and 5 TFs, which outperformed differentially expressed miRNAs or TFs in the ccRCC prognosis. How many molecules out of the eleven prognostic factors have been used in current commercial molecular panels for pan-cancer or ccRCC diagnosis? What are the functional connections between these selected miRNAs and TFs?

Response 3: Thank you for your comments. At present, among 11 prognostic molecules, KLF5, IRF1, MYC, IKZF1, hsa-miR-144-5p, hsa-miR-223-3p and hsa-miR-183-5p have been reported to serve as biomarker in cancer [3-10], but only the MYC is acted as a commercial molecular biomarker in metastatic breast cancer [11]. Currently, no commercial molecular is applied to ccRCC diagnosis.

In this work, the functional connections between these selected miRNAs and TFs may be that, on the one hand, down-regulated expressed TFs could down-regulate some miRNA expression to further up-regulate tumor suppressor gene expression, whilst the down-regulated miRNAs could also up-regulate TF expression to further up-regulate the expression of tumor suppressor miRNA to down-regulate the expression of some oncogenes; on the other hand, up-regulated TFs could up-regulate miRNA expression to inhibite the expression of some tumor suppressor genes, whilst the up-regulated miRNAs could also down-regulate the expression of some TFs to further down-regulate the expression of some miRNAs to up-regulate the expression of some tumor suppressor genes. Although these connections have rarely been reported in ccRCC, some of them have been reported in other cancers. For example, KLF5 could promote the expression of miR-145, miR-124 and miR-183 by binding to their promoters to involve in the invasive pituitary adenoma [12]. The homologous proteins TFAP2C of TFAP2A can also promote lung tumorigenesis and aggressiveness through miR-183 and miR-33a-mediated cell cycle regulation [13]. The interaction of MYC and hsa-miR-144 could involve in chronic myelogenous leukemia cell K562 [14]. These studies might support the interaction between transcription factors and miRNAs involved in the development of ccRCC. These above detailed descriptions have been added in the discussion sections in this revised manuscript.

Point 4: Page 14 Figure 11, it is very interesting mechanism map. However, discussion of the proposed mechanism in a more specified way (i.e. which miRNAs, TFs, target genes were involved? how were they involved? and how did they interplay with each other, and so on) is desired.

Response 4: Your comment is a really good point. Here, we have discussed it according to your comments, and the new content has been added into the revised manuscript. For example, as shown in this Figure 10 of the revised manuscript, down-regulated transcription factors such as KLF5 and GATA2 can down-regulated miR-183-3p and miR-1269a to up-regulated FAS and other tumour suppressor genes, whilst down-regulated miR-183-3p and miR-1269a can also up-regulated IKZF1 and IRF1 to down-regulated downstream targets. Additionally, up-regulated MYC and IKZF1 can activate miR-223-3p and miR-365b-3p to down-regulate RPS6KA6, DEPTOR and other tumour suppressor genes, whilst miR-223-3p and miR-365b-3p can also down-regulated TFAP2A and GATA2 to down-regulate miR-183-5p and miR-1269a to up-regulated FAS and VEGFA and so on.

Point 5: Generally, the authors need to reorganize the manuscript to make it focused, concise, and easy to read. BTW, I suggest the author to reconsider the article title.

Response 5: Thanks for your wise advice. We have modified the manuscript title. The new title is “Transcription factor and miRNA interplays can manifest the survival of ccRCC pateins”.

Reference

Gao, Q.; Zhu, H.; Dong, L.; Shi, W.; Chen, R.; Song, Z.; Huang, C.; Li, J.; Dong, X.; Zhou, Y., et al. Integrated Proteogenomic Characterization of HBV-Related Hepatocellular Carcinoma. Cell 2019, 179, 561-577 e522, doi:10.1016/j.cell.2019.08.052. Cancer Genome Atlas Research, N. Comprehensive molecular characterization of clear cell renal cell carcinoma. Nature 2013, 499, 43-49, doi:10.1038/nature12222. Mishra, P.J. MicroRNAs as promising biomarkers in cancer diagnostics. Biomarker research 2014, 2, 19, doi:10.1186/2050-7771-2-19. Braga, E.A.; Fridman, M.V.; Loginov, V.I.; Dmitriev, A.A.; Morozov, S.G. Molecular Mechanisms in Clear Cell Renal Cell Carcinoma: Role of miRNAs and Hypermethylated miRNA Genes in Crucial Oncogenic Pathways and Processes. Frontiers in genetics 2019, 10, 320, doi:10.3389/fgene.2019.00320. Yang, N.; Ekanem, N.R.; Sakyi, C.A.; Ray, S.D. Hepatocellular carcinoma and microRNA: new perspectives on therapeutics and diagnostics. Advanced drug delivery reviews 2015, 81, 62-74, doi:10.1016/j.addr.2014.10.029. Trop-Steinberg, S.; Azar, Y. Is Myc an Important Biomarker? Myc Expression in Immune Disorders and Cancer. The American journal of the medical sciences 2018, 355, 67-75, doi:10.1016/j.amjms.2017.06.007. Walch-Ruckheim, B.; Pahne-Zeppenfeld, J.; Fischbach, J.; Wickenhauser, C.; Horn, L.C.; Tharun, L.; Buttner, R.; Mallmann, P.; Stern, P.; Kim, Y.J., et al. STAT3/IRF1 Pathway Activation Sensitizes Cervical Cancer Cells to Chemotherapeutic Drugs. Cancer research 2016, 76, 3872-3883, doi:10.1158/0008-5472.CAN-14-1306. Marrero-Rodriguez, D.; Taniguchi-Ponciano, K.; Jimenez-Vega, F.; Romero-Morelos, P.; Mendoza-Rodriguez, M.; Mantilla, A.; Rodriguez-Esquivel, M.; Hernandez, D.; Hernandez, A.; Gomez-Gutierrez, G., et al. Kruppel-like factor 5 as potential molecular marker in cervical cancer and the KLF family profile expression. Tumour biology : the journal of the International Society for Oncodevelopmental Biology and Medicine 2014, 35, 11399-11407, doi:10.1007/s13277-014-2380-4. Fang, Q.; Zhao, X.; Li, Q.; Li, Y.; Liu, K.; Tang, K.; Wang, Y.; Liu, B.; Wang, M.; Xing, H., et al. IKZF1 alterations and expression of CRLF2 predict prognosis in adult Chinese patients with B-cell precursor acute lymphoblastic leukemia. Leukemia & lymphoma 2017, 58, 127-137, doi:10.1080/10428194.2016.1180682. Larne, O.; Ostling, P.; Haflidadottir, B.S.; Hagman, Z.; Aakula, A.; Kohonen, P.; Kallioniemi, O.; Edsjo, A.; Bjartell, A.; Lilja, H., et al. miR-183 in prostate cancer cells positively regulates synthesis and serum levels of prostate-specific antigen. European urology 2015, 68, 581-588, doi:10.1016/j.eururo.2014.12.025. Santa-Maria, C.A.; Kruse, M.; Raska, P.; Weiss, M.; Swoboda, A.; Mutonga, M.B.; Abraham, J.; Jain, S.; Nanda, R.; Montero, A.J. Impact of tissue-based genomic profiling on clinical decision making in the management of patients with metastatic breast cancer at academic centers. Breast cancer research and treatment 2017, 166, 179-184, doi:10.1007/s10549-017-4415-1. Yang, W.; Xu, T.; Qiu, P.; Xu, G. Caveolin-1 promotes pituitary adenoma cells migration and invasion by regulating the interaction between EGR1 and KLF5. Experimental cell research 2018, 367, 7-14, doi:10.1016/j.yexcr.2018.01.008. Kang, J.; Kim, W.; Lee, S.; Kwon, D.; Chun, J.; Son, B.; Kim, E.; Lee, J.M.; Youn, H.; Youn, B. TFAP2C promotes lung tumorigenesis and aggressiveness through miR-183- and miR-33a-mediated cell cycle regulation. Oncogene 2017, 36, 1585-1596, doi:10.1038/onc.2016.328. Liu, L.; Wang, S.; Chen, R.; Wu, Y.; Zhang, B.; Huang, S.; Zhang, J.; Xiao, F.; Wang, M.; Liang, Y. Myc induced miR-144/451 contributes to the acquired imatinib resistance in chronic myelogenous leukemia cell K562. Biochemical and biophysical research communications 2012, 425, 368-373, doi:10.1016/j.bbrc.2012.07.098.

Reviewer 2 Report

To further screen new reliable miRNAs as diagnostic and prognostic biomarkers, we here firstly screened eight potentially prognostic miRNAs based on RNA-seq and clinical information from the TCGA database-What is the exact criteria for choosing potentially prognostic and filtering it down to only 8?

This result showed that the prognostic efficiency and credibility of the eight-miRNA signature significantly outperformed a single miRNA, implying that miRNAs synergistical regulation play key roles in the tumorigenesis and progression of ccRCC-How it was interpreted in terms of scoring and probability?

Additionally, we found that twenty-two TFs could interact with eight miRNAs based on deepCAGE, TransmiR v2.0 and MirWalk3.0 database- Individual data set or consensus data was considered?

We further constructed a interplay network of TFs and miRNAs, and the network analysis revealed that the interplay between twenty-two TFs and eight miRNAs could control synergistically the expression of oncogenes, driver genes and tumor suppressor genes-parameters for network analysis with justification

The molecular mechanism of the interplay between transcription factors and miRNAs improving the prognosis of ccRCC patients-Highlight the part that focuses on novel understanding.

Present work is predictive analysis. Relevant clinical studies are required to consolidate the role of novel prognostic miRNAs.

Otherwise work is well presented and explained.
